# Cells Responding to Closely Related Cholesterol-Dependent Cytolysins Release Extracellular Vesicles with a Common Proteomic Content Including Membrane Repair Proteins

**DOI:** 10.3390/toxins15010004

**Published:** 2022-12-20

**Authors:** Sara Alves, Joana M. Pereira, Rupert L. Mayer, Alexandre D. A. Gonçalves, Francis Impens, Didier Cabanes, Sandra Sousa

**Affiliations:** 1Cell Biology of Bacterial Infections, IBMC, i3S—Instituto de Investigação e Inovação em Saúde, Universidade do Porto, 4200-135 Porto, Portugal; 2Molecular and Cellular (MC) Biology PhD Program, ICBAS—Instituto de Ciências Biomédicas Abel Salazar, University of Porto, 4050-313 Porto, Portugal; 3VIB-UGent Center for Medical Biotechnology, VIB, 9052 Ghent, Belgium; 4Department of Biomolecular Medicine, Ghent University, 9052 Ghent, Belgium; 5VIB Proteomics Core, VIB, 9052 Ghent, Belgium; 6Molecular Microbiology, IBMC, i3S—Instituto de Investigação e Inovação em Saúde, Universidade do Porto, 4200-135 Porto, Portugal

**Keywords:** pore-forming toxins, plasma membrane repair, shedding, extracellular vesicles, listeriolysin O, pneumolysin, calcium influx, proteomics

## Abstract

The plasma membrane (PM) protects cells from extracellular threats and supports cellular homeostasis. Some pathogens produce pore-forming toxins (PFTs) that disrupt PM integrity by forming transmembrane pores. High PFT concentrations cause massive damage leading to cell death and facilitating infection. Sub-lytic PFT doses activate repair mechanisms to restore PM integrity, support cell survival and limit disease. Shedding of extracellular vesicles (EVs) has been proposed as a key mechanism to eliminate PFT pores and restore PM integrity. We show here that cholesterol-dependent cytolysins (CDCs), a specific family of PFTs, are at least partially eliminated through EVs release, and we hypothesize that proteins important for PM repair might be included in EVs shed by cells during repair. To identify new PM repair proteins, we collected EVs released by cells challenged with sub-lytic doses of two different bacterial CDCs, listeriolysin O and pneumolysin, and determined the EV proteomic repertoire by LC-MS/MS. Intoxicated cells release similar EVs irrespectively of the CDC used. Also, they release more and larger EVs than non-intoxicated cells. A cluster of 70 proteins including calcium-binding proteins, molecular chaperones, cytoskeletal, scaffold and membrane trafficking proteins, was detected enriched in EVs collected from intoxicated cells. While some of these proteins have well-characterized roles in repair, the involvement of others requires further study. As proof of concept, we show here that Copine-1 and Copine-3, proteins abundantly detected in EVs released by intoxicated cells, are required for efficient repair of CDC-induced PM damage. Additionally, we reveal here new proteins potentially involved in PM repair and give new insights into common mechanisms and machinery engaged by cells in response to PM damage.

## 1. Introduction

The integrity of the plasma membrane (PM) is crucial for the survival of single cells and maintenance of tissue organization in a full organism. PM lesions cause massive Ca^2+^ influx, loss of cytoplasmic content and disruption of cellular homeostasis, and lead to cell death if rapid reseal does not occur [1]. Therefore, to survive PM damage, cells promptly engage repair mechanisms that aim to restore PM integrity and cellular homeostasis. In physiological conditions, PM lesions and repair often occur in tissues under mechanical stress, such as skeletal and cardiac muscle, intestinal and respiratory epithelia and vascular endothelium [2]. Deregulation of the PM repair pathways was largely found to be associated with the development of different human disorders [2,3]. Infection by microbial pathogens and exacerbated immune responses also cause PM damage that ultimately determines cell fate and disease progression [4,5].

Bacterial pore-forming toxins (PFTs) are major virulence factors produced by a variety of severe human pathogens [6]. They are secreted by the bacteria as soluble monomers, interact with the PM of host cells and assemble into stable transmembrane pores [7] to disrupt epithelial barriers, support immune evasion and promote infection dissemination [4,8]. Cholesterol-dependent cytolysins (CDCs) are well conserved, reaching at least 45% of structural homology and sharing similar mechanisms of pore formation as the most studied subfamily of PFTs [9]. Monomers of CDCs directly bind PM cholesterol residues, oligomerize and undergo conformational changes that allow their stable insertion across the PM, forming a large well-defined pore including around 40 monomers [10,11]. The toxin concentration critically determines the level of the induced PM damage, which in turn dictates cell fate [12]. While cells are unable to cope with high toxin concentrations engaging in cell death pathways, at sub-lytic concentrations of PFTs, the cells orchestrate a multifactorial repair response that supports cell survival and limits the progression of the infection [4,12,13].

Distinct mechanisms, including protein clogging, wound patching, membrane replacement by endocytosis or shedding of damaged sites, were proposed to participate in the re-establishment of PM integrity [2,4]. However, their hierarchy in effective repair and the level of interdependency and/or cooperation remains unknown. The proper activation of repair mechanisms relies on the controlled influx of extracellular Ca^2+^ [14]. Fluctuations in intracellular Ca^2+^ concentrations are efficiently sensed by Ca^2+^-binding proteins, such as annexins, that are activated and recruited to the sites of damage to participate in the PM reseal [15]. Additionally, Ca^2+^ influx causes the disruption and reorganization of the cortical cytoskeleton and fosters PM remodeling events promoting blebbing and shedding of extracellular vesicles, which were both previously correlated with PM damage resolution [13,16,17]. PM blebs were suggested to confine the pore away from the cytosol, avoiding further Ca^2+^ increase and thus protecting the cell cytosol from irreversible damage [18]. Such blebs can either retract or be released as extracellular vesicles allowing pore elimination and/or playing immunomodulatory functions [18,19,20,21,22].

Previous studies demonstrated that proteins involved in PM repair regulate the dynamics of PM blebbing and shedding of extracellular vesicles upon intoxication [13]. These data lead us to hypothesize that, upon pore formation, increased intracellular Ca^2+^ levels induce the recruitment of the cellular machinery for repair to the cortex of the cells next to the sites of pore insertion. Thus, such machinery might control PM remodeling and shedding of vesicles to assist pore elimination. We thus propose that shed vesicles might carry at least part of the cellular machinery required for PM repair. Here, we seek to determine the proteomic repertoire of vesicles released by intoxicated cells during the repair of PM damage, with the aim of identifying novel proteins involved in repair. To further assess whether repair mechanisms were conserved among damage induced by different CDCs, we used two different toxins and performed a comparative analysis. Purified listeriolysin O (LLO) and pneumolysin (PLY), which are produced by the pathogenic gram-positive bacteria *Listeria monocytogenes* and *Streptococcus pneumoniae*, respectively, were used [12,23]. A previous study determined the proteomic content of vesicles released by cells intoxicated with PLY [21]. Here, we seek to advance on this research and perform a comparative analysis revealing the proteins that are common in the response to different CDCs and specific to the release of extracellular vesicles triggered by such toxins.

We reveal here new protein candidates potentially involved in the repair of PM damage induced by CDCs. Additionally, our data open new perspectives and provide new insights into common mechanisms and machinery engaged by cells to respond to a broad range of PM damage.

## 2. Results and Discussion

### 2.1. The Concentration of CDCs Determines the Extent of PM Damage and Efficacy of Repair

We first set up the experimental conditions in which we cause the maximum PM damage that triggers the maximum capacity to repair and recover PM integrity, assuming that this correlates with maximal release of extracellular vesicles. We started by determining the highest concentration of CDCs that allow the full recovery of PM integrity following toxin washout. HeLa cells were incubated with increasing concentrations of purified LLO or PLY for 15 min and allowed to recover from induced PM damage for 20 h. PM permeability was monitored immediately after toxin washout (0 h) and 20 h of recovery by propidium iodide (PI) permeability assays coupled with flow cytometry analysis. We observed that the percentage of PI-positive cells increased in response to growing concentrations of LLO and PLY (gray bars, Figure 1A,B), thus confirming that both LLO and PLY permeabilize the PM in a dose-dependent manner [13,21,24]. Specifically, 20 h after toxin washout, to allow recovery from PM damage, the percentage of PI-positive cells dropped to the levels detected in non-intoxicated (NI) cells (blue bars, Figure 1A,B) in samples challenged with lower doses of toxins, thus indicating that these cells are able to fully recover PM integrity. At the highest CDC concentrations (2 nM LLO and 0.5 nM PLY), at least 80% of the cells displayed PI permeability after toxin washout (0 h, gray bars, Figure 1A,B); and after 20 h of recovery, the percentage of PI-positive cells remained significantly higher than in the NI samples (blue bars, Figure 1A,B), suggesting that at such CDC concentrations cells do not recover from inflicted damage.

To ensure that the decreased percentage of PI-positive cells after 20 h of recovery was indeed due to PM repair and not related to the loss of damaged cells, we quantified by microscopy the number of adherent cells for each sample. In NI control conditions, the number of adherent cells increased after 20 h, which corresponds to normal cell growth (Figure 1C,D). In samples intoxicated with low CDCs doses, the number of adherent cells remained roughly the same immediately after toxin washout (0 h) and recovery (20 h), suggesting that cells intoxicated with lower doses of CDCs recover PM integrity, do not die but do not proliferate (Figure 1C,D). At the highest CDC concentrations (2 nM LLO and 0.5 nM PLY), the number of adherent cells (at 0 and 20 h) was significantly lower than in the NI condition (Figure 1C,D), suggesting that at such concentrations the cells undergo immediate overwhelming damage, are unable to repair the damage and recover PM integrity, and detach from the support (Figure 1C,D).

Together, these data indicate that at sub-lytic CDC concentrations cells become permeabilized but do not massively detach and thus are not lost during the experiment. While at high CDC doses, the cells detach and are lost; and thus, the percentage of PI-positive cells is likely underestimated in our experimental setup. Expectedly, our results also indicate that the level of PM permeabilization depends on the concentration and activity of CDCs and ability of cells to repair the injury decreases as damage increases. Recovery of PM damage is achieved in a cell population in which about 60% of the cells are permeabilized. The concentration and activity of CDCs are thus key determinants for the effectiveness of repair.

Given that we aim to cause the maximum damage that cells are able to cope with, in the next experiments we will use 1 nM for LLO and 0.2 nM for PLY, concentrations that permeabilize roughly 60% of the cells and allow complete recovery as compared to the NI condition (Figure 1A,B).

### 2.2. Repair of CDC-Induced PM Damage Relies on an Abrupt Shot of Ca^2+^ Early in Intoxication

In addition to the concentration and associated activity of the toxin, the concentration of extracellular Ca^2+^ is another key factor determining the repair and recovery of PM integrity [4]. Of note, under physiological conditions, Ca^2+^ exists at ~100 nM in the cell’s cytosol and may reach ~2 mM in the extracellular milieu [25]. To determine the extracellular Ca^2+^ concentration that would allow maximum repair of CDC-induced damage, HeLa cells were challenged with 1 nM LLO for 15 min in the presence of the physiological concentration of extracellular Ca^2+^ (1.2 mM) and left to recover from damage, after toxin washout, for 20 h in Hanks’ Balanced Salt Solution (HBSS) supplemented with growing Ca^2+^ concentrations. PM integrity was assessed by PI permeability assays immediately after toxin washout (0 h) and after 20 h of recovery in Ca^2+^ concentrations ranging from 0 to 1.2 mM. We found that cells do not recover PM integrity in the absence of extracellular Ca^2+^; notably, in such conditions, the level of damage even increased during the 20 h, as indicated by the increased percentage of cells permeable to PI (Figure 2A). However, in the presence of extracellular Ca^2+^, cells recovered their PM integrity at similar levels irrespective of the Ca^2+^ concentration that we tested (Figure 2A). These results indicate that a low concentration of extracellular Ca^2+^ (75 µM) is necessary and sufficient to enable PM repair, and that higher concentrations of extracellular Ca^2+^ have no significant effect on PM repair. We next investigated whether Ca^2+^ is only important during the repair stage or also during the intoxication. In particular, we aimed to assess if the availability of extracellular Ca^2+^ during the repair process would overcome the damage caused in the absence of Ca^2+^. For that, HeLa cells were intoxicated with 1 nM LLO for 15 min in the absence of extracellular Ca^2+^ and allowed to recover from damage for 20 h in HBSS supplemented with growing concentrations of Ca^2+^. As above, PM integrity was assessed by PI permeability assays after toxin washout (0 h) and 20 h of recovery. As expected, samples intoxicated in the absence of Ca^2+^ showed higher percentages of PI-positive cells as compared with cells intoxicated in the presence of Ca^2+^ (Figure 2A,B). This indicates that in the absence of Ca^2+^, cells undergo increased PM damage. Additionally, we found that cells intoxicated in the absence of extracellular Ca^2+^ do not recover PM integrity, even if physiological Ca^2+^ concentrations are provided during the recovery period (Figure 2B). Indeed, the percentages of PI permeable cells remained unchanged after 20 h of recovery (Figure 2B). Altogether these results suggest that during exposure to CDCs, extracellular Ca^2+^ and its consequent rapid influx at early stages of intoxication determine the ability of damaged cells to reseal their PM. If intoxication occurs in the absence of extracellular Ca^2+^, the later addition of Ca^2+^ cannot promote repair and recovery of PM integrity. The early entry of Ca^2+^ upon pore formation likely activates mechanisms of repair that cannot be triggered at later stages. Our data are in agreement with previous studies reporting that in vitro chelation of extracellular Ca^2+^ during PLY intoxication leads to a twofold reduction in the production and shedding of EVs [19,21] and also results in inefficient Ca^2+^-dependent repair [26].

In this context, we investigated whether different extracellular Ca^2+^ concentrations impact CDC-induced PM damage in the initial intoxication stages. For that, HeLa cells were incubated with 1 nM LLO for 15 min, in the presence of Ca^2+^ concentrations ranging from 0 to 1.2 mM, and PM permeability was immediately assessed. Intoxications performed in concentrations of Ca^2+^ below 1.2 mM resulted in equivalent percentages of PI-positive cells (Figure 2C), which were significantly higher than those obtained for intoxications under physiological Ca^2+^ concentrations (1.2 mM). Surprisingly, at 0.1 mM of Ca^2+^, which corresponds to 10-fold less than the physiological extracellular Ca^2+^ concentration, but that is still 1000-fold higher than the intracellular concentration; the damage caused by CDCs is already too high and likely irreversible (Figure 2C). These data suggest that repair of PM damage induced by CDCs relies on an at least 12,000-fold gradient maintained by cells between intracellular and extracellular Ca^2+^ concentrations, which causes an abrupt entry of Ca^2+^ at early stages upon pore formation. Our results thus indicate that levels of Ca^2+^ under the physiological concentrations during the intoxication have a profound impact on the ability of cells to recover PM integrity.

We concluded that to obtain the maximum recovery of PM integrity upon CDC pore formation, both the intoxication and recovery time periods should be performed in physiological Ca^2+^ concentrations (1.2 mM); otherwise, PM damage increases and reaches levels that are incompatible with full PM recovery. Also, given the similarities between LLO and PLY [11,27], we decided to use the same Ca^2+^ concentrations in intoxications with either LLO or PLY.

### 2.3. Cells Intoxicated with CDCs Release Large Extracellular Vesicles during PM Damage Repair

As mentioned above, vesicle shedding is a major mechanism deployed by cells to repair PM damage [4]. To characterize shed vesicles during repair of LLO- and PLY-induced PM damage, HeLa cells were intoxicated in the conditions determined above (1 nM LLO or 0.2 nM PLY for 15 min in the presence of 1.2 mM Ca^2+^) and allowed to recover from damage in the presence of 1.2 mM Ca^2+^. After 3 h of repair, which we found sufficient for full recovery (Appendix A), shed vesicles were isolated from cell supernatants by ultracentrifugation. As the control, we collected vesicles shed by the same number of NI cells during 3 h. The concentration and size distribution of purified extracellular vesicles (EVs) were determined by Nanoparticle Tracking Analysis (NTA). We found that, during recovery, LLO- and PLY-intoxicated cells released similar amounts of vesicles (Figure 3A–C). Additionally, samples from supernatants of intoxicated cells contained a two-fold higher EV concentration than samples from control NI cells (Figure 3B). This confirms that increased EV shedding is a common mechanism responding to CDC-induced damage, activated during the recovery of PM integrity. Size distribution analysis revealed that while the concentration of small-size EVs (<120 nm) was equivalent in the supernatants of intoxicated and NI cells (Figure 3C), larger EVs (120 nm to 500 nm) were shed in significantly increased amounts by intoxicated cells as compared with NI control cells (Figure 3C). Transmission electron microscopy (TEM) analysis of purified EVs confirmed that those recovered from supernatants of LLO- and PLY-intoxicated cells displayed a heterogeneous profile with sizes ranging from 120 to 500 nm, while EVs from NI cells showed a homogeneous profile with sizes up to 120 nm (Figure 3D).

These data indicate that vesicle shedding is increased during repair of CDC-induced damage and that intoxicated cells mainly release large EVs, which are rarely released by NI cells. Furthermore, LLO- and PLY-intoxicated cells release similar amounts of vesicles, which share the same size distribution profile, reinforcing the idea that vesicle shedding is a common response triggered by different CDCs to promote PM repair.

### 2.4. The Release of Large Extracellular Vesicles Contributes to the Removal of CDCs from the PM

We further confirm the concept that shedding of EVs constitutes a mechanism for the physical removal of the pore from the PM, thus allowing CDC elimination and recovery of PM integrity [21,28,29,30]. To ascertain if EVs collected from supernatants of CDC-intoxicated cells contain the CDCs, we performed immunogold labeling on purified EVs using specific antibodies raised against LLO or recognizing GFP (to detect GFP-PLY). Electron microscopy images displayed in Figure 4A,B, respectively, show the presence of LLO or PLY associated with EVs released during recovery of damage. The percentage of EVs associated with gold-labeled particles was determined per field in TEM images. EVs associated with at least one gold particle were considered positive for LLO or PLY. We found that the majority of vesicles released are positive for either LLO or PLY (Figure 4C). However, LLO and PLY were undetectable in about 40% of the EVs, suggesting that vesicle shedding may serve different purposes and that cells may also release vesicles in areas where the PM is intact. LLO and PLY were also detected in purified EVs by immunoblotting (Figure 4D) and mass spectrometry (MS)-based proteome analysis (Figure 4E) on vesicles shed by LLO- and PLY-intoxicated cells, respectively. As expected, CDCs were not detected in EVs released by NI cells.

Altogether these data confirm that shedding of EVs allows the cells to get rid of PM-bound CDCs, as previously proposed [29,30,31]. Additionally, these results strongly suggest that shedding has an active role in pore elimination and protects the cell from damage supporting cell survival. Nevertheless, the absence of CDCs in 40% of the EVs also raises the possibility that released EVs might have additional functions such as transport of the repair machinery to the cell cortex, PM reorganization and/or delivery of signals to neighboring cells. In line with this, a previous study had proposed that PLY triggers the release of distinct vesicle populations: PLY-positive and PLY-negative EVs, which would supposedly be derived from different regions of the PM [21]. One may, therefore, speculate that shed EVs originate from different routes and have distinct functions during PM repair, which may also correlate with the different size sub-populations.

### 2.5. Proteins Present in EVs Released by Intoxicated Cells Are Potential Actors in PM Repair

To gain further information on the function of EVs during the repair of PM damage induced by CDCs, we determined their protein content by MS-based proteomics. As a control, we used EVs released by NI HeLa cells in two different conditions: EVs were collected from cells cultured in standard conditions (10% EV-depleted FBS, control) for 72 h and cells cultured under serum starvation (1% EV-depleted FBS, control-starved) for 48 h. Serum starvation was expected to act as a stress stimulus for EVs release [32]. As NI cells shed a lesser amount of EVs than intoxicated cells, the time before collection of control EVs was extended to ensure that we had enough material for analysis. Considering that donor cells can specifically pack into EVs the cellular components required to respond to a specific condition, we postulated that proteins overrepresented in EVs released by intoxicated cells might potentially be involved in repair or intercellular signaling responses.

EVs collected from three independent experiments were lysed, proteins were digested by trypsin and peptide mixtures were separated and analyzed by liquid chromatography-tandem mass spectrometry (LC-MS/MS). A total of 2110 proteins were reliably quantified (Appendix A). Principal component analysis (PCA), a statistical method used to reduce the number of variables from a dataset to produce strong patterns, clearly demonstrated that EVs from NI HeLa cells cluster differently from EVs released by intoxicated cells (Figure 5A). This suggests that proteins are selectively packed in EVs released by intoxicated cells, further supporting the idea that the identification of the proteins enriched in EVs is an invaluable step toward the holistic understanding of PM repair mechanisms. Additionally, together with the differences in EV concentration and size distribution (Figure 3), our results suggest that the release of EVs triggered by CDCs is a specific response to damage different from canonical EV secretion. Moreover, EVs from LLO- or PLY-intoxicated cells cluster together indicating that they share similar proteins (Figure 5A), corroborating the idea that different CDCs trigger equivalent cell protective responses.

To identify the proteins overrepresented in EVs released by intoxicated cells, which we postulate might be involved in PM repair, several comparative analyses of the proteomic data were performed. Volcano plots were generated for pairwise comparison of protein intensities in the different conditions: control vs. control-starved, LLO vs. PLY, LLO vs. control; PLY vs. control and all CDCs (LLO and PLY, mentioned as PFTs) vs. all controls. The control vs. control-starved comparison revealed 50 significantly regulated proteins (though 41 of these are potential contaminants, Volcano plot in Appendix A), while comparing LLO vs. PLY did not show any significant differences (Volcano plot in Appendix A). Together, these comparisons confirm that EVs from control cells have similar proteomic profiles and EVs released by LLO- or PLY-intoxicated cells share the same proteomic signature, corroborating the PCA analysis (Figure 5A). Notably, the LLO vs. control comparison revealed 360 significantly regulated (including 57 potential contaminants), of which 46 were overrepresented in LLO samples, including LLO itself (Figure 4E, Appendix A). Similarly, the PLY vs. control comparison resulted in 102 regulated proteins (including 48 potential contaminants), of which 13 were overrepresented in PLY samples, including PLY itself (Figure 4E, Appendix A). Of note, the potential contaminants are defined by the software tool that we have used, which includes a list of common contaminants in proteomic analysis.

Given the reported similarities between the modes of action of LLO and PLY and associated cellular responses [4,7,33], as well as the closely related proteomic signature of the EVs released by LLO- or PLY-intoxicated cells (Figure 5A), we also compared protein intensities in all EVs samples from intoxicated cells (by either LLO or PLY) with the intensities in all control samples. This comparison revealed 615 proteins with significantly different abundances, though 45 of these are potential contaminants (Figure 5B, Appendix A). The intensity of these proteins was also visualized in a heatmap, clustering in three distinct groups (Figure 5C). Cluster 1 comprises 70 proteins that are overrepresented in the EVs collected from supernatants of intoxicated cells (Figure 5C). Proteins included in this cluster, which according to our assumption may be key for the repair of PM damage, were classified following Gene Ontology analysis (Figure 5D and Table 1) using PANTHER [34,35,36]. Calcium-binding proteins, molecular chaperones, cell adhesion molecules, cytoskeletal, scaffold and membrane trafficking proteins were found among the most enriched categories in the EVs released by intoxicated cells (Figure 5D and Table 1). In particular, we detected several annexins (Table 1), which have been extensively described as key players in Ca^2+^-driven repair responses [21,26,37,38,39]. Chaperones and cytoskeletal proteins were also previously reported to be essential for effective repair of damage caused by CDCs [13,16]. The evidence that proteins with well-established roles in PM repair were detected enriched in the EVs released by intoxicated cells validates our approach. Our data further reveal other EV-enriched proteins that might be important for the repair of damage induced by CDCs. Finally, results also suggest that repair relies on complex cellular responses that require the cooperation of a variety of protein families. The specific function of newly detected candidates in the response to CDCs and infection needs to be further assessed in proper cellular and animal models.

### 2.6. Copine-1 and Copine-3 Are Required for Efficient PM Repair

To validate the proteomic data, we investigated whether Copine-1 and Copine-3 (CPNE1 and CPNE3, respectively) were important for the repair of PM damage induced by PLY. Some observations supported a role for CPNE1 and CPNE3 in the response to CDCs: although copine’s exact function in cells remains unknown, CPNE1 and CPNE3 are ubiquitously expressed in several tissues [40] and were described as calcium-dependent phospholipid-binding proteins [41,42]. Also, they were both found enriched in EVs released from intoxicated cells (Table 1), along with other proteins with established functions in PM repair. We generated HeLa cell lines depleted for CPNE1 or CPNE3 by CRISPR-Cas9 genome editing and investigated their ability to repair PLY-induced PM damage. The expression of CPNE1 and CPNE3 in the generated cell lines was verified by Western blot using specific antibodies recognizing as either CPNE1 or CPNE3 (Appendix A). We found that the cell line deficient in CPNE1 (KO-CPNE1) was notably not expressing CPNE1 but expressing CPNE3 as their wild-type (WT) parental cells. However, the cell line knockout for CPNE3 (KO-CPNE3) fails to express either CPNE3 or CPNE1, which suggests that the selected guide RNAs might be targeting both proteins or that in the absence of CPNE3, CPNE1 is less expressed or degraded. Both cell lines, together with HeLa WT cells, were challenged with 0.2 nM of PLY, and their PM integrity was assessed by PI permeability assays immediately after toxin washout (0 h) and at different time points (30, 60, and 120 min) after toxin washout and recovery in complete medium. At 0 h, immediately after toxin washout, the three cell lines were permeabilized at comparable levels, about 70% of the cells were detected as PI-positive (Figure 6A). In HeLa WT cells, the PM permeability rapidly decreased to the levels of NI cells. Specifically, 30 min after toxin washout only 20% of the cells were detected as PI-positive, indicating that these cells efficiently recover from PM damage induced by sub-lytic doses of PLY (Figure 6A). Contrarily, intoxicated KO-CPNE1 and KO-CPNE3 HeLa cells displayed percentages of PI-positive cells significantly higher than NI cells throughout the recovery period (Figure 6A), indicating that in the absence of CPNE1 and/or CPNE3 cells do not efficiently repair the injury inflicted by PLY. Of note, the repair defect appears to be amplified in KO-CPNE3 as compared with CPNE1 cells (Figure 6A), which may be related to the fact that KO-CPNE3 cells lack both CPNE1 and CPNE3. These data point out that CPNE1 and CPNE3 are required for efficient repair of damage induced by PLY and reinforce our hypothesis suggesting that the EVs released from intoxicated cells carry into their cargo intracellular machinery that is specifically recruited to damaged areas and plays a pivotal role in the re-establishment of the PM integrity.

Considering that CPNE1 and CPNE3 translocate to the PM in response to increased intracellular Ca^2+^ concentrations, we investigated the intracellular localization of both CPNE1 and CPNE3 upon incubation of cells with PLY. For that, HeLa cells were transfected to express GFP-tagged CPNE1 or CPNE3 and the cellular localization of the GFP protein fusions was assessed by confocal microscopy in non-intoxicated cells or upon the 0.2 nM PLY challenge. In non-intoxicated cells, the GFP-associated fluorescence was homogenously detected in the cytoplasm, indicating that both GFP-CPNE1 and GFP-CPNE3 are uniformly distributed in the cytoplasm (Figure 6B). However, upon PLY intoxication, GFP-CPNE1 and GFP-CPNE3 were found enriched at the cell cortex, suggesting that the Ca^2+^ influx generated by PLY pore formation induces the translocation of CPNE1 and CPNE3 to PM-damaged sites (Figure 6B). Additionally, in intoxicated cells, CPNE1 and CPNE3 were found to accumulate together with NMHCIIA (Figure 6C) at cortical sites where actomyosin cytoskeleton is remodeling to support effective repair [13,16]. Considering that accumulations of NMHCIIA in the cortex of intoxicated cells are established as hallmarks of PM damage repair, the detection of CPNE1 and CPNE3 at these specific sites strongly suggests their role in assisting repair.

Together, these data point to the important role of CPNE1 and CPNE3 in the recovery of PM integrity possibly through the interaction with the repair machinery at specific cortical sites. Importantly, these results serve as proof-of-concept supporting our original hypothesis proposing that proteins present in the EVs released by intoxicated cells are potentially required for damage repair.

## 3. Conclusions

PM shedding is considered a major repair mechanism in response to several sources of PM injury, including laser, mechanical, detergent, and PFT-induced damage [13,21,28,29,30,43,44]. Our data demonstrate that shedding of EVs is more than a stochastic event that only serves to eliminate the damaged PM. Notably, in addition to allowing the physical removal of CDCs from the cell surface [21,30,31], the release of EVs may serve as a communication vehicle and a transporter of repair proteins. Importantly, both CDC-positive and CDC-negative EVs were detected, raising the hypothesis that these two populations may have different functions, and accordingly the CDC might not be the only difference in their proteomic profile. Also, given that the proteomic profile of EVs released by NI and intoxicated cells is substantially different, it is worth speculating that EVs released upon PM damage follow secretion routes that differ from canonical EVs, with a cargo that might be stimulus-specific.

Here, we provide the full protein repertoire of EVs released in the context of intoxication, and supply a high value list of proteins that are potentially new actors in PM repair of CDC-induced damage, as well as in the response to other types of PM damage. In particular, EVs released by intoxicated cells appear to be enriched in calcium-binding proteins, molecular chaperones, cell adhesion molecules; and cytoskeletal, scaffold and membrane trafficking proteins. Of note, a previous study analyzed the proteins enriched in microvesicles released by PLY intoxicated cells using as a control the total membrane isolates from intoxicated cells [21]. Despite the identification of some overlapping proteins (e.g., annexins), we also found differences in the identified proteins, which may be related to the different controls used.

Among the proteins of interest identified here are CPNE1 and CPNE3, AHNAK and sorcin. As mentioned, CPNE1 and CPNE3 are phospholipid-binding proteins that respond to intracellular Ca^2+^ increases, thereby translocating from the cytoplasm to the cell cortex [41,42]. Additionally, there is increasing evidence suggesting that copines may mediate signal transduction [45,46] and membrane transport, namely by interacting with annexins [47,48]. Our data here reveal CPNE1 and CPNE3 as critical components for an efficient PM repair in response to PLY-induced damage; however, their specific molecular role in this process needs to be further studied. In particular, CPNE1 and CPNE3 protein partners during repair need to be identified. Additionally, whether CPNEs are required for the repair of other types of PM damage is still unknown. The neuroblast differentiation-associated protein AHNAK was previously associated with actin cytoskeleton remodeling and membrane repair through interactions with annexins [49,50]. Notably, AHNAK was found in vesicles that are rapidly exocytosed and recycled back in response to Ca^2+^ increase [51,52]. Finally, sorcin, which is also a Ca^2+^-binding protein, was reported to be involved in vesicular trafficking and co-localizes with Rab11 [53], a vesicular trafficking protein required for PM repair and pore removal of a non-CDC PFT [54].

Additionally, some chromatin-associated proteins (essentially histones) and translational proteins (ribosomal proteins) were unexpectedly detected enriched in EVs. However, their presence in EVs does not necessarily indicate that they are involved in repair and may have signaling functions that need to be investigated. Several histones were modified in response to different CDCs to regulate gene expression [55,56]. Noteworthy, histones were detected in EVs released in other contexts, and it was speculated that they may act as damage-associated molecular pattern signals controlling cell signaling and innate immunity [57]. Concerning the translational proteins, it is possible that such proteins are packed together with specific RNA molecules, which together with proteins are commonly included in EVs. Ribosome-associated proteins were often detected in the EVs released in other contexts [58,59]. The presence of RNA in EVs released by intoxicated cells and their association with ribosomal proteins needs to be further investigated.

Our work reveals the proteomic content of EVs released by cells upon bacterial CDCs attack, confirms the role of these EVs in the elimination of the CDC pore and identifies new proteins involved in PM repair, giving new insights into the cell machinery engaged in response to PM damage. Further studies are now required to assess the exact molecular role of these proteins in PM repair.

## 4. Materials and Methods

### 4.1. Cell Lines and Reagents Culture

HeLa (ATCC CCL-2) and HEK293T (ATCC CRL-3216) cells (ATCC, Manassas, VA, USA) were cultured in Dulbecco’s Modified Eagle’s Medium (DMEM) with glucose and L-glutamine (Lonza, Basel, Switzerland) and supplemented with 10% fetal bovine serum (FBS; Biowest, Nuaillé, France). Cells were maintained at 37 °C in a humidified 5% CO_2_ atmosphere.

### 4.2. Cell Intoxications

LLO and PLY toxins were purified as described [60]. Intoxications were performed in Hanks’ balanced salt solution (HBSS) (Lonza, Basel, Switzerland) for 15 min, at the indicated toxin concentrations. For recovery assays, the toxins were washed out twice with PBS Ca^2+^-free (Lonza, Basel, Switzerland) and allowed to recover from damage in DMEM supplemented with 10% FBS for 20 h. For experiments aiming to assess the importance of extracellular Ca^2+^ concentration, intoxications and recovery assays were carried out in HBSS Ca^2+^-free medium and supplemented with the indicated Ca^2+^ concentrations.

### 4.3. Flow Cytometry Analysis

Plasma membrane integrity was assessed by flow cytometry, following propidium iodide (PI) permeability assays. HeLa cells (2.5 × 10^5^) were seeded in six-well plates 20 h before intoxicated as indicated and allowed to recover from damage. Immediately after toxin washout (T 0 h) and 20 h of recovery, cells were trypsinized and resuspended in 0.5 mL of PBS containing 2% of FBS. Samples were then incubated with PI at 2 µg/mL for 1 min and fluorescence intensity was measured at the FL3 channel on a BD Accuri™ C6 flow cytometer (BD Biosciences, San Jose, CA, USA). At least 10,000 cells were collected after doublet discrimination and data was analyzed using FlowJo Software (Version 10.8 TreeStar Inc., Ashland, OR, USA).

For the analysis of HeLa WT, KO CPNE1 and KO CPNE3 cell lines, 3 × 10^5^ cells were seeded in 6-well plates 24 h before intoxication with 0.2 nM of PLY. PI incorporation was measured as described above immediately after toxin washout (T 0 h) and after recovery during the indicated time points At least 16,000 cells were collected and analyzed.

### 4.4. Immunofluorescence Microscopy Quantifications

Cells intoxicated as described in Section 4.2, were fixed with 4% paraformaldehyde (PFA; 15713, EMS, Hatfield, PA, USA) for at least 10 min at room temperature, quenched with 0.1 M NH_4_Cl for 40 min, permeabilized with 0.1% Triton-X100 in PBS for 5 min and washed 3× in PBS. Coverslips were incubated for 30 min with Phalloidin Alexa Fluor 488 (Invitrogen, Waltham, MA, USA) and DNA was stained with DAPI (Sigma, St Louis, MO, USA). Coverslips were mounted onto microscope slides with Aqua-Poly/Mount (Polysciences, Warrington, PA, USA). Images were collected with an epifluorescence Olympus BX63 (Olympus, Tokyo, Japan) microscope equipped with a 20 × 0.75 NA objective lens. DAPI nuclear staining was used to quantify the number of cells after washout (0 h) and 20 h of recovery after intoxication. The number of cells/field was quantified in blind in 5 random fields for each condition from three independent experiments.

### 4.5. Isolation of Extracellular Vesicles (EVs)

HeLa cells (3 × 10^6^) were seeded in 100 mm tissue culture dishes and incubated in DMEM supplemented with 10% EVs-depleted FBS. Then, 20 h later, cells were left non-intoxicated (NI) in HBSS or intoxicated with 1 nM of LLO, 0.2 nM of PLY or GFP-PLY (for immunogold labeling assays) during 15 min. NI and intoxicated cells were then washed twice in PBS Ca^2+^-free and allowed to recover from damage in serum-free DMEM for 3 h. Cell culture supernatant was collected from 20 tissue culture dishes and EVs were isolated by differential centrifugation as described in [61] with slight modifications. Briefly, cell culture supernatants were collected and centrifuged at 2000× *g* for 20 min at 4 °C to remove cell debris, transferred to polypropylene tubes and centrifuged at 30,000× *g* for 40 min at 4 °C. The resulting EV pellet was washed in PBS, centrifuged again at 30,000× *g* (40 min at 4 °C) and resuspended in 70 µL of PBS. Samples were immediately processed for transmission electron microscopy (TEM) experiments or stored at 4 °C for 24 h for immunogold labeling and LC-MS/MS analysis. For longtime storage, EVs were kept at −80 °C until further processing.

### 4.6. Nanoparticle Tracking Assay

Size distribution profiles of isolated EVs were obtained using the NanoSight NS300 system (Malvern Panalytical, Worcestershire, UK). Diluted vesicle suspensions were loaded into the instrument and 30 s videos were recorded in triplicates for each sample, with camera level set at 15 and detection threshold set at 5. The videos were analyzed with the NTA 3.2 software (Malvern Panalytical, Worcestershire, UK) to assess size distribution.

### 4.7. Electron Microscopy and Immunogold Labeling

The morphology of EVs was assessed using negative staining TEM. Briefly, formvar-carbon coated EM-grids were glow discharged for 20 s at 10 mA. 5 µL of EV sample was added to the grid and allowed to adhere for 5 min. EVs were then stained with 7 µL of 1% uranyl acetate, excess of liquid was dropped onto a piece of filter paper and grids were allowed to dry prior to imaging. For immunogold labeling, EVs samples were fixed in 2% PFA for at least 10 min at 4 °C. Glow discharged formvar-carbon coated EM-grids were floated on top of 10 µL drops of fixed EVs for 20 min, washed in PBS and incubated with PBS 0.02% glycine for 10 min, to quench free aldehyde groups. Grids were washed in PBS, incubated in PBS 2% BSA containing 0.1% saponin for 30 min, washed in PBS 0.5% BSA and incubated with PBS 1% BSA containing primary antibody mouse anti-LLO (1:5, antibody B8B20-3-2 [62]) or rabbit anti-GFP (1:10, antibody Ia1 produced in our lab) for 2 h 30 min at room temperature. The grids were washed three times in PBS 0.5% BSA then incubated with a goat anti-mouse IgG conjugated to 15 nm gold (1:20, EM GMHL15/2, BBI-Solutions OEM Limited, Cardiff, UK) or anti-rabbit IgG conjugated to 10 nm gold (1:20; Sigma, St Louis, MO, USA) for 1 h at room temperature. The grids were washed three times in PBS 0.5% BSA, incubated with 1% glutaraldehyde for 5 min and washed three times in deionized water prior to negative staining. Images were acquired on a JEOL JEM 1400 TEM (JEOL, Tokyo, Japan) 80 kV equipped with a CCD digital camera Orious 1100 W. Digital images of EVs from each sample were taken from 3 independent experiments.

### 4.8. Proteomics Sample Preparation and LC-MS/MS Analysis

LLO- and PLY-induced EVs were isolated from the supernatant of intoxicated HeLa cells allowed to recover for 3 h in serum-free medium. Control EVs were isolated from the supernatant of NI HeLa cells cultured in DMEM supplemented with 10% EV-depleted FBS (control) or 1% EV-depleted FBS (control-starved), for 72 h or 48 h, respectively. Proteins were solubilized with 100 mM Tris pH 8.5, 1% sodium deoxycholate, 10 mM tris(2-carboxyethyl) phosphine (TCEP) and 40 mM chloroacetamide for 10 min at 95 °C under 1000 rpm (Thermomixer, Eppendorf, Hamburg, Germany). Each sample was processed for proteomics analysis following the solid-phase-enhanced sample-preparation (SP3) protocol as described in [63]. Enzymatic digestion was performed with 2 µg of Trypsin/LysC overnight at 37 °C under 1000 rpm. Tryptic sample peptides were dissolved in 0.1% formic acid and 500 ng per sample injected for LC-MS/MS analysis into an Ultimate 3000 RSLCnano system. Trapping was performed at 10 μL/min for 3 min in loading solvent (0.1% formic acid) on a 0.5 cm trapping cartridge (Acclaim PepMap C18 100 Å, 5 mm × 0.3 mm, 160454, Thermo Scientific, Waltham, MA, USA). Peptide separation after trapping was performed on a 50 cm EASY-Spray column (ES803, PepMap RSLC, C18, 2 μm, Thermo Scientific, Waltham, MA, USA) at 35 °C.

Peptides were eluted by a non-linear gradient from 2.5 to 50% MS solvent B (0.1% FA 80% ACN) over 148 min, at a constant flow rate of 250 nL/min, followed by a 15 min washing at 99% MS solvent B. Re-equilibration with 97.5% MS solvent A (0.1% FA in water) was performed at 250 nL/min for 17 min adding up to a total run length of 180 min. Peptides were analyzed by a data-dependent top10 acquisition method in positive ion mode on a Q Exactive Orbitrap instrument (Thermo Scientific, Waltham, MA, USA). MS1 scans were acquired at *m*/*z* 380–1580 with an MS1 AGC target of 3E6 ions with a max. ion injection time of 60 ms at a resolution of 70,000. MS2 scans were acquired at resolution 35,000 with an AGC target of 2E5 with 110 ms max. injection time, 2Da isolation window, intensity threshold 7.3E4, exclusion of unassigned, 1, 8, >8 positively charged precursors, peptide match preferred, exclude isotopes on, dynamic exclusion time 45 s. The scan range for MS2 spectra was set from 200 to 2000 *m*/*z*. HCD energy was set to 27% NCE, and the background ion at 445.12003 Da was used as lock mass. The spray voltage was set to 1.9 kV.

### 4.9. Proteomics Data Analysis

LC-MS/MS runs of all 12 biological samples were searched together combining technical replicates for statistical reasons using the MaxQuant algorithm (created by Jürgen Cox, version 1.6.9.0, https://www.maxquant.org/, accessed on 9 November 2022, Max-Plank Institute of Biochemistry, Martinsried, Germany). Default search settings were used, including a false discovery rate set at 1% on PSM and protein level. Spectra were searched against the human protein sequences in the Swiss-Prot database (database release version of 2018_11), containing 20,412 sequences [64], supplemented with the LLO and PLY sequences (P13128 and Q04IN8, respectively) available at UniProt database. The mass tolerance for precursor and fragment ions was set to 4.5 and 20 ppm, respectively, during the main search. Enzyme specificity was set to the C-terminal of arginine and lysine, also allowing cleavage next to prolines with a maximum of two missed cleavages. Variable modifications were set to oxidation of methionine residues and acetylation of protein N-termini. Matching between runs was enabled with a matching time window of 0.7 min and an alignment time window of 20 min. Only proteins with at least one unique or razor peptide were retained, leading to the identification of 4585 protein groups. Proteins were quantified by the MaxLFQ algorithm integrated into the MaxQuant software. A minimum ratio count of two unique or razor peptides was required for quantification. Further data analysis was performed with the Perseus software (version 1.6.7.0) after uploading the protein groups file from MaxQuant [65]. Reverse database hits were removed, and replicate samples were grouped. Proteins with less than three valid values in at least one group were removed, and missing values were imputed from a normal distribution around the detection limit resulting in 2110 quantified proteins, which were subsequently used for further data analysis. These quantified proteins were subjected to two-sided, unpaired *t*-tests using permutation-based multiparameter correction with 1000 randomizations and an FDR of 1% plotting eVs from LLO-treated cells against control eVs, eVs from PLY-treated cells against control eVs, and eVs from all toxin-treated cells (PLY and LLO, mentioned as PFTs) against all control eVs (control and control-starved). The result of the latter *t*-test is shown in the volcano plots in Figure 5B. For each protein, the log2 (PFT/controls) fold change value is indicated on the X-axis, whereas the statistical significance (−log *p*-value) is indicated on the Y-axis. Proteins outside the curved lines, set by an FDR value of 0.05 and an S0 value of 1 in the Perseus software, represent specific significant upregulated or downregulated proteins. The 615 proteins identified with statistically different abundances between toxin-treated eVs were taken further in Perseus for z-scoring of the LFQ intensities and hierarchical clustering, resulting in the heat maps presented in Figure 5C.

### 4.10. Establishment of CPNE1 and CPNE3 Knockout HeLa Cells by CRISPR/Cas9

Oligonucleotides targeting the gene encoding for CPNE1 and CPNE3 were designed using Benchling [66]. Two independent sequences were designed for each gene, targeting exons 4 and 10 of CPNE1 and exons 4 and 13 of CPNE3 (pre-gRNA listed in Appendix A). Forward and reverse sequences were annealed and cloned in pLentiCRISPR V2 plasmid gift from Feng Zhang; Addgene plasmid #52961) [67] using the *Bsm*BI site. HEK293T cells were transfected to assemble and release lentivirus expressing the gRNAs targeting CPNE1 and CPNE3 encoding genes. Briefly, transfections were performed using Lipofectamine^TM^ 2000 (Invitrogen, Waltham, MA, USA) diluted in OptiMEM (Thermo Scientific, Waltham, MA, USA) and mixed with 0.5 µg of plasmid p8.91 (gift from Simon Davis; Addgene plasmid #187441); 0.5 µg of plasmid pMDG (gift from Simon Davis; Addgene plasmid #187440) [68] and 0.25 µg of pLentiCRISPR V2 harboring the selected gRNAs. Lentivirus was recovered 72 h later from the supernatants of transfected HEK293T cells. HeLa cells were seeded at 1 × 10^6^ cells per well in 6-well plates and incubated at 37 °C in a humidified atmosphere with 5% CO_2_. When 50% confluency was reached, they were incubated with the recovered lentivirus and selected with 2.5 µg/mL of puromycin.

### 4.11. Generation of GFP-Tagged CPNE1 and CPNE3

DNA fragments encoding CPNE1 and CPNE3 were amplified by PCR from the cDNA of HeLa cells, using the high fidelity ACCUZYME^TM^ DNA polymerase (Bioline, London, UK) and the primers listed in Appendix A. The generated PCR fragments were cloned into the plasmid pEGFP-C1 (Clontech, Mountain View, CA, USA). The CPNE1 encoding fragment was inserted using the restriction sites *Xho*I and *Sal*I (New England BioLabs, Ipswich, MA, USA), and the CPNE3 fragment was cloned into *Xho*I and *Bam*HI (New England BioLabs, Ipswich, MA, USA) restriction sites. The generated plasmids allowed the expression of GFP-tagged CPNE1 or CPNE3 variants, in which the GFP is placed at the N-terminus of the fusion protein (GFP-CPNE1 or GFP-CPNE3). The constructs were verified by sequencing.

### 4.12. Transfection and Immunofluorescence Microscopy

HeLa cells (3 × 10^5^) were seeded on glass coverslips in 6-well plates and transfected 20 h after with 3 μg of plasmid encoding GFP-CPNE1 or GFP-CPNE3 using the jetPRIME^®^ transfection reagent (Polyplus, Illkirch, France) following the manufacturer’s instructions. Cells were left non-intoxicated or incubated with 0.2 nM PLY for 10 min, fixed with 4% PFA for at least 10 min at room temperature, quenched with 0.1 M NH_4_Cl for 40 min at 4 °C, permeabilized with 0.1% Triton-X100 in PBS for 5 min, and washed 3× in PBS. Afterward, coverslips were incubated for 1 h with rabbit anti-NMHCIIA (Sigma, St Louis, MO, USA) followed by 45 min with a goat anti-rabbit coupled to CY3 (Jackson ImmunoResearch, West Grove, PA, USA). DNA was stained with DAPI (Sigma, St Louis, MO, USA). Coverslips were mounted onto microscope slides with Aqua-Poly/Mount (Polysciences, Warrington, PA, USA). Images were acquired with a Leica Scanning Confocal SP5 equipped with a 63 × 1.30 NA Glycerol objective lens and processed using Image J [69].

### 4.13. Immunoblotting Assay

HeLa cells or EVs were directly lysed in Laemmli buffer (0.25 mM Tris-HCl pH 6.8; 10% SDS; 50% glycerol; and 5% β-mercaptoethanol). Total protein extracts were resolved by SDS-PAGE and transferred onto nitrocellulose membranes (Hybond ECL; GE Healthcare, Chicago, IL, USA). Primary antibodies (mouse anti-LLO (1:100, antibody B8B20-3-2 [62]), rabbit anti-GFP (1:100, antibody Ia1 produced in our lab), mouse anti-Copine 1 (1:250, sc-101269 Santa Cruz Biotechnology, Dallas, TX, USA), mouse anti-Copine 3 (1:500, sc-390143, Santa Cruz Biotechnology, Dallas, TX, USA) and anti-actin (1:5000, AC15 Sigma Aldrich, St Louis, MO, USA) and secondary antibodies (anti-mouse HRP or anti-rabbit HRP 1:1000, Abliance, Compiègne, France) were diluted in TBS-Tween (150 mM NaCl; 50 mM Tris-HCl, pH 7.4; and 0.1% Tween). A signal was detected using ECL (Thermo Scientific, Waltham, MA, USA).

### 4.14. Statistical Analysis

Statistical analyses were performed using Prism GraphPad 6 software (GraphPad Software, La Jolla, CA, USA, www.graphpad.com, accessed on 2 January 2019). Two-way ANOVA with Dunnett’s post-hoc analyses was used to compare the means of samples in relation to a control sample, within each condition. One-way ANOVA with Tukey’s post-hoc analyses was used for pairwise comparisons of more than two different means. Two-tailed unpaired Student’s *t*-test was used for the comparison of means between two samples. Details regarding the statistical analysis of proteomic data are provided in the corresponding figure legends and/or methods sections above.

## Figures and Tables

**Figure 1 toxins-15-00004-f001:**
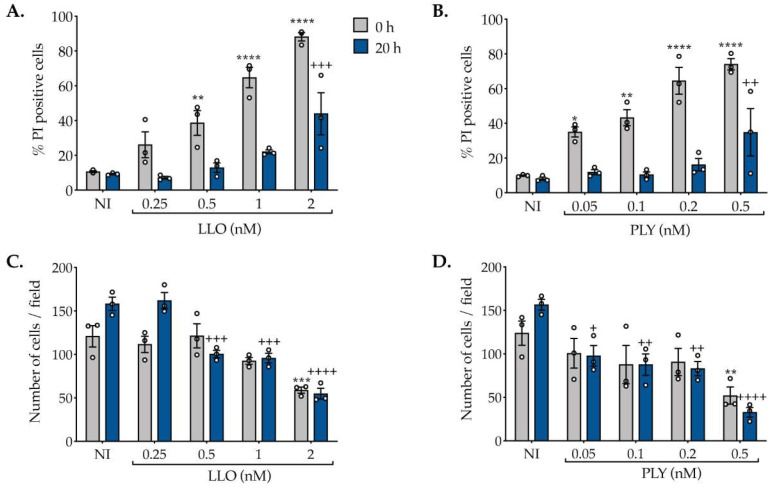
CDC concentration determines the extent of PM damage and the efficacy of repair. HeLa cells were left non-intoxicated (NI) or were intoxicated with the indicated concentrations of (**A**) LLO or (**B**) PLY for 15 min. PM permeability was measured by flow cytometry following incorporation of the membrane impermeable dye propidium iodide (PI). The graphs show the percentage of PI-positive cells immediately after toxin washout (0 h, gray bars) or following toxin washout and recovery (20 h, blue bars). (**C**,**D**) Quantification of the total number of cells per field (5 fields corresponding to at least 250 cells were quantified per experiment in each condition) after DAPI nuclear staining. HeLa cells were left NI or were intoxicated with the indicated concentrations of (**C**) LLO or (**D**) PLY for 15 min. Quantifications were performed immediately after toxin washout (0 h, gray bars) or following toxin washout and recovery (20 h, blue bars). Values represent the mean ± SEM of three independent experiments. Open circles represent values for each independent experiment. *p* values were calculated using two-way ANOVA followed by Dunnett’s test. ^+^
*p* < 0.05; ** and ^++^
*p* < 0.01; *** and ^+++^
*p* < 0.001; **** and ^++++^
*p* < 0.0001. * refers to NI at 0 h vs. intoxicated at 0 h; + refers to NI at 20 h vs. intoxicated at 20 h.

**Figure 2 toxins-15-00004-f002:**
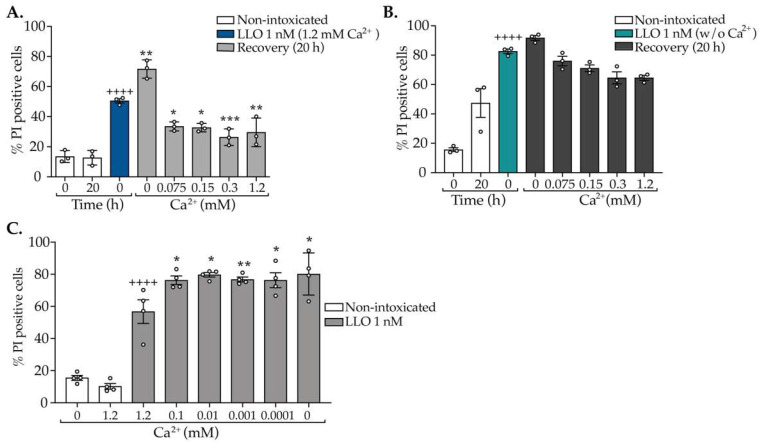
Physiologic concentrations of extracellular Ca^2+^ are required for the resealing of damaged PM. HeLa cells were left non-intoxicated (NI) or were intoxicated for 15 min with 1 nM of LLO in (**A**) HBSS containing 1.2 mM Ca^2+^ or in (**B**) HBSS without Ca^2+^. (**A**,**B**) After toxin washout, cells were allowed to recover PM integrity for 20 h in HBSS supplemented with increasing Ca^2+^ concentrations. Graphs (**A**,**B**) show the percentage of PI-positive cells determined by flow cytometry at the end of intoxication after LLO washout (0 h) and after 20 h of recovery in HBSS with the indicated Ca^2+^ concentrations. (**C**) HeLa cells were left NI or intoxicated with 1 nM LLO in HBSS with the indicated Ca^2+^ concentrations. The percentage of PI-positive cells was assessed immediately after washout. Values are the mean ± SEM of four independent experiments. Open circles represent the value for individual experiments. *p* values were calculated using one-way ANOVA followed by Tukey’s test. * *p* < 0.05; ** *p* < 0.01; *** *p* < 0.001; ^++++^
*p* < 0.0001. * refers to intoxicated cells in the presence of 1.2 mM Ca^2+^ at 0 h vs. recovery in growing Ca^2+^ concentrations (in **A**) and vs. cells intoxicated in the presence of Ca^2+^ concentrations ranging from 0–0.1 mM (in **C**); + symbols refer to NI at 0 h vs. intoxicated cells at 0 h (in **A**–**C**).

**Figure 3 toxins-15-00004-f003:**
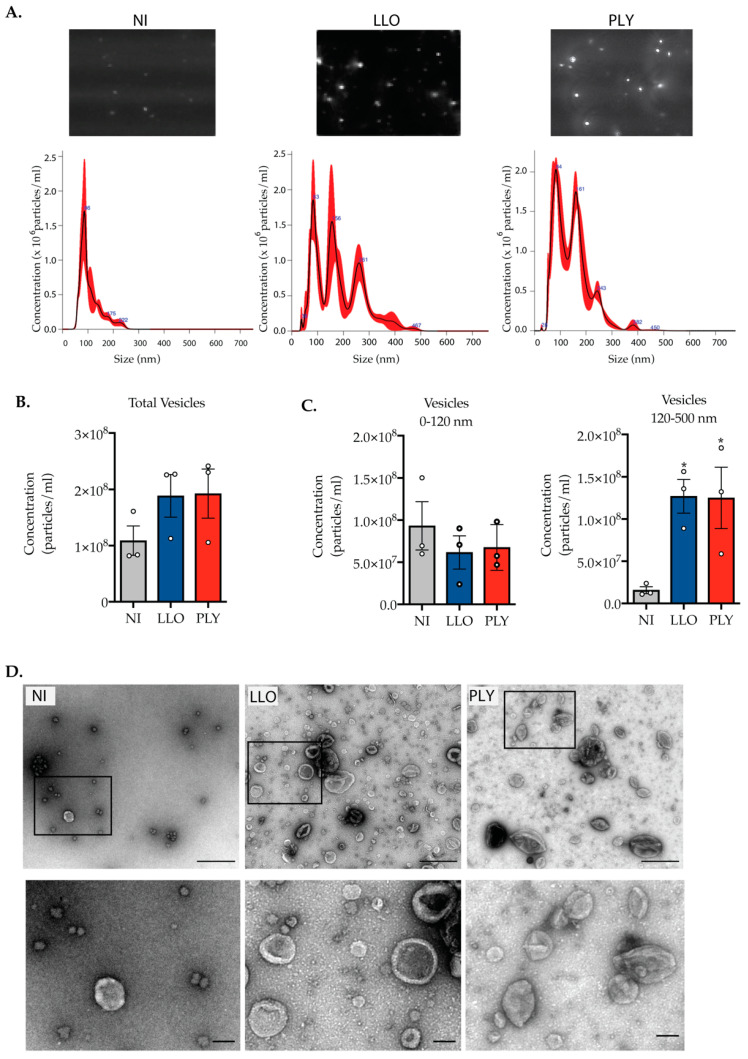
Release of large extracellular vesicles is a specific response to intoxication by CDCs. (**A**–**C**) Nanoparticle tracking analysis (NTA) of purified EVs isolated from the cell supernatants of HeLa cells non-intoxicated (NI) or intoxicated with 1 nM of LLO or 0.2 nM of PLY, for 15 min. (**A**) Light scattering snapshots from videos recorded from NanoSight analysis (upper panel). Representative histograms obtained from NTA analysis showing, for each sample, the concentration of vesicles according to their size (bottom panel). (**B**) The graph shows the total concentration of EVs recovered in each sample. Values are the mean ± SEM of three independent experiments. (**C**) Size distribution of quantified EVs. Graphs show the concentration of vesicles with sizes between 0 to 120 nm and between 120 to 500 nm. Values are the mean ± SEM of three independent experiments. Open circles represent the value for individual experiments. A one-way ANOVA test was performed (* *p* < 0.05). (**D**) Transmission electron microscopy (TEM) of purified EVs. Scale bar = 500 nm. Insets show EVs at higher magnification (Scale bar = 100 nm).

**Figure 4 toxins-15-00004-f004:**
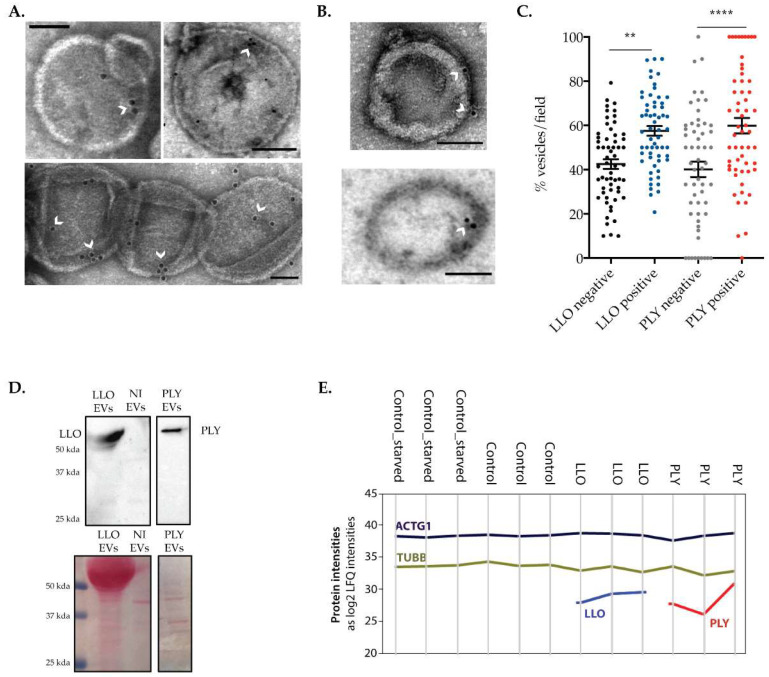
The release of EVs contributes to the removal of CDCs from the PM of host cells. (**A**,**B**) Transmission electron microscopy (TEM) images showing immunogold labeling of (**A**) LLO in EVs collected from supernatants of HeLa cells intoxicated with 1 nM of LLO for 15 min and (**B**) GFP-PLY in EVs collected from supernatants of HeLa cells intoxicated with 0.2 nM of GFP-PLY for 15 min. Arrowheads indicate gold particles in the EVs collected from LLO- and PLY-intoxicated cells. Scale bar, 100 nm. (**C**) Quantification of LLO- and PLY-positive EVs in TEM images. Evs associated at least with one gold particle were considered positive. Data from three independent experiments are shown as percentage of CDC-positive or -negative vesicles per field. A one-way ANOVA test was performed (** *p* < 0.01, **** *p* < 0.0001). (**D**) Immunoblot showing LLO and PLY protein levels in EVs released from HeLa upon intoxication. EVs released by non-intoxicated (NI) cells were used as the control. Ponceau staining (lower panel) was used as loading control to detect proteins in membranes before immunoblotting. (**E**) Profile plots showing protein levels (as log2 LFQ intensities) of PLY, LLO, as well as TUBB and ACTG1 as loading controls. All LFQ intensities based on match-between-runs with missing MS/MS scans were removed prior to visualization. Detection of LLO and PLY in EVs of the respective CDC-treated cells confirms release into EVs.

**Figure 5 toxins-15-00004-f005:**
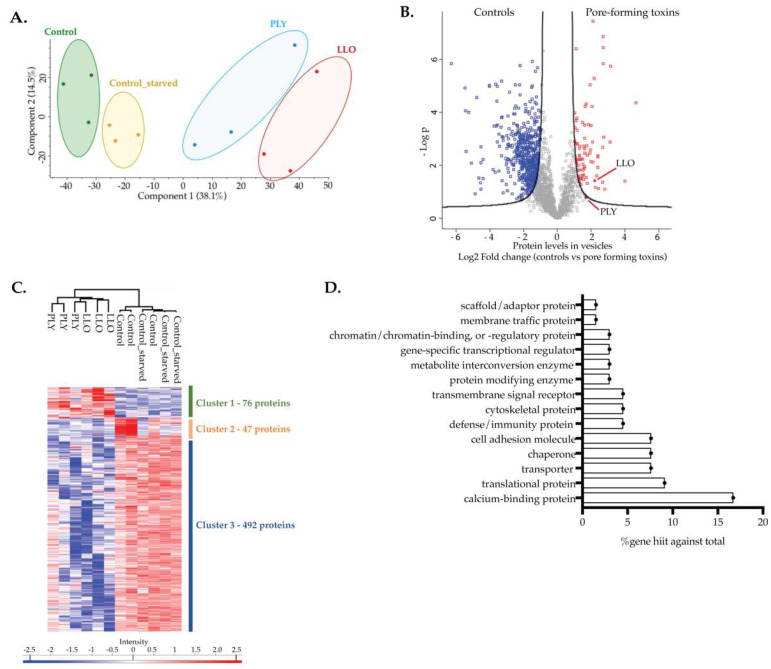
CDC-damaged cells release vesicles with specific proteomic profiles enriched in proteins involved in PM repair. (**A**) Principal component analysis (PCA) based on all proteins detected in EVs released from HeLa cells non-intoxicated (NI; control and control-starved) and intoxicated with LLO or PLY. Scatter plot shows sample mappings along the two principal components. X and Y axis show principal component 1 and principal component 2, which explain 38.1% and 14.5% of the total variance, respectively. While both principal components represent a dimensionality reduction of all protein abundances, they consider different protein abundances to a different extent. (**B**) Volcano plot showing the protein level represented in fold change (log2) of EVs released from NI HeLa cells (controls) compared to EVs collected from HeLa cells intoxicated with LLO and PLY (X axis). Replicate samples from three independent experiments were analyzed and Student *t*-tests were applied to calculate −log *p*-values for each protein (Y axis). The curved black lines represent the threshold for statistical significance determined by *t*-testing (FDR = 0.05 and S0 = 1). Blue squares correspond to underrepresented proteins and red squares are overrepresented proteins. (**C**) Heatmap visualizing the protein intensity of significant hits from (**B**) after non-supervised hierarchical clustering. (**D**) Gene ontology analysis of proteins from cluster 1 in (**C**), performed on PANTHER.

**Figure 6 toxins-15-00004-f006:**
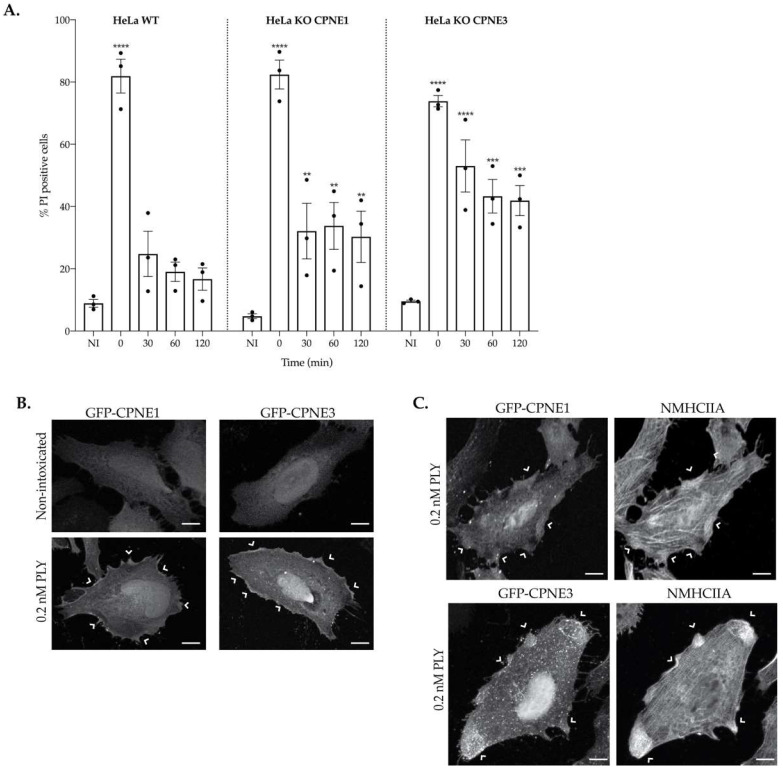
CPNE1 and CPNE3 are required for efficient PM repair of damage induced by PLY. (**A**) WT, KO-CPNE1 and KO-CPNE3 HeLa cells were left non-intoxicated (NI) or were intoxicated with 0.2 nM of PLY (10 min) followed by toxin washout and recovery allowed for 30, 60 or 120 min. Permeability of the PM was determined by flow cytometry following PI incorporation at the indicated time points. The graph shows quantifications of PI-positive cells for all cell lines in non-intoxicated (NI) or intoxicated conditions. Values are the mean ± SEM (n = 3) and *p*-values were calculated using two-way ANOVA followed by Dunnett’s test. ** *p* < 0.01; *** *p* < 0.001; **** *p* < 0.0001; statistics refer to NI vs. intoxicated with different times of recovery. (**B**) Representative confocal microscopy images of HeLa cells overexpressing GFP-CPNE1 or GFP-CPNE3 left non-intoxicated or challenged with 0.2 nM PLY for 10 min. GFP signal is shown. Arrows indicate GFP-CPNE1 and GFP-CPNE3 enrichments at the cortex of PLY intoxicated cells. (**C**) Confocal microscopy images of HeLa cells overexpressing GFP-CPNE1 or GFP-CPNE3 intoxicated with 0.2 nM PLY (10 min) and stained for NMHCIIA. GFP-CPNEs and NMHCIIA signals are shown. Arrows show the co-accumulations of GFP-CPNEs and NMHCIIA at sites of actomyosin remodeling for repair. Scale bar, 10 µm.

**Table 1 toxins-15-00004-t001:** Full list of proteins detected in cluster 1, classified following gene ontology analysis. All these proteins were found statistically enriched in the EVs released by CDC-damaged cells.

Cluster 1	
**Calcium binding proteins** Annexin A1 Annexin A2 Annexin A3 Annexin A4 Annexin A5 Annexin A6 Annexin A7 Annexin A11 Calmodulin-like protein 5 Copine-1 Copine-3 Hippocalcin-like protein 1 **Translational proteins** 40S ribosomal protein S20 40S ribosomal protein S18 60S ribosomal protein L8 60S ribosomal protein L31 60S ribosomal protein L23a Eukaryotic translation initiation factor 4B **Transporters** ATP synthase subunit G2, mitochondrial Cationic amino acid transporter 2 Monocarboxylate transporter 4 Sodium- and chloride-dependent taurine transporter Solute carrier family 12 member 2 **Chaperones** DnaJ homolog subfamily A member 1 Peptidyl-prolyl cis-trans isomerase B Proteasome assembly chaperone 1 Protein disulfide-isomerase A3 **Cell adhesion molecules** Cadherin-2 Cadherin-13 C-X-C chemokine receptor type 4 Desmocollin-1 Podocalyxin **Chromatin-binding, or -regulatory proteins** Histone H2A.V Histone H2B type 1-L Histone H3.3 Histone H4 **Scaffold/adaptor proteins** A-kinase anchor protein 12	**Cytoskeletal proteins** Keratin, type I cytoskeletal 18 Vimentin PDZ and LIM domain protein 5 **Defense/immunity proteins** HLA class I histocompatibility antigen, B alpha chain HLA class I histocompatibility antigen, C alpha chain UL16-binding protein 3 **Transmembrane signal receptors** CD44 antigen Ephrin type-A receptor 2 Tissue factor **Protein modifying enzymes** Protein kinase C alpha type Multifunctional procollagen lysine hydroxylase and glycosyltransferase LH3 **Membrane trafficking proteins** Alpha-taxilin **Gene-specific transcriptional regulators** Glucocorticoid receptor Leucine-rich repeat flightless-interacting protein 1 **Metabolite interconversion enzymes** 5-nucleotidase Carnitine O-palmitoyltransferase 1, liver isoform **Others** Proteasomal ubiquitin receptor ADRM1 Glycogen synthase, muscle CD59 glycoprotein Neuroblast differentiation-associated protein AHNAK Neuroplastin Brain acid soluble protein 1 CD99 antigen Nuclease-sensitive element-binding protein 1 Y-box-binding protein 3 Myristoylated alanine-rich-C-kinase substrate Sorcin Ubiquitin-40S ribosomal protein S27a Dermcidin RNA-binding protein 14 78 kDa glucose-regulated protein RNA-binding protein 4 Uncharacterized protein C7orf50

## Data Availability

The mass spectrometry proteomics data have been deposited to the ProteomeXchange Database (ProteomeXchange Datasets) via the PRIDE [70] partner repository with the dataset identifier PXD037166 (reviewer access with username: reviewer_pxd037166@ebi.ac.uk and password: waZnat8o).

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
