# Peer review of "Cells Responding to Closely Related Cholesterol-Dependent Cytolysins Release Extracellular Vesicles with a Common Proteomic Content Including Membrane Repair Proteins"

_toxins, 2022, doi:10.3390/toxins15010004_

Round 1

Reviewer 1 Report

In the manuscript: “Cells responding to different bacterial pore-forming toxins release extracellular vesicles with a common proteomic content including membrane repair proteins.” the main conclusion of the authors`` is: “We show here that indeed PFTs are at least partially eliminated through EVs release and hypothesize that proteins important for PM repair might be included in EVs shed by cells during repair”.

However, data supporting that microvesicle dependent PM repair mechanisms after PLY, SLO and LLO stimulation were published already from year 2015 to 2021 and thus shown in HEK293T cells and immune cell types (Data of the current manuscript is based on HEK293T and HeLa cells). The authors` and the editor may want to check it by themselves:  PMID: 27481675, PMID: 29979630, PMID: 27481675, PMID: 33572185, and PMID: 31914676.

Major Concerns

This paper is a duplication of already published work from 2020 by Larpin et al. (PMID: 31914676), without even referring to this paper and from Wolfmeier et al. BBA 2016, which is cited but out of context. Please compare Figures from Larpin et al, FASEB 2020 to the submitted manuscript. In fact, the authors` use the same experimental strategies, i.e. PI influx assays, Nanosight NTA, EM, and toxin-microvesicle protein analysis to duplicate data with another cell type or even the same cell type (used in Wolfmeier et al. 2016), without referring why they have chosen to use HeLa and HEK293T cell and what the focus of the study was. The published proteomic composition of PLY induced microvesicles from HEK293T cells is published in Wolfmeier et al., 2016 (PMID: 27481675) and proteins identified from the current study do not provide new information.

Thus, the manuscript is completely lacking novelty, duplicates published experimental data, and lacks proper citation of previous published works in the field, that already elaborated a model of PM repair through microvesicle shedding form HEK293T and other cell types.

This manuscript is not suitable to be published in Toxins or anywhere else in its current form.

Author Response

REVIEWER 1

We deeply regret the opinion of the reviewer on our manuscript. We are obviously aware of the published studies she/he mentions and cited some of those reports in a way that seemed fair and appropriate. Nevertheless, we added a small paragraph in the revised version of the manuscript, summing up the mentioned reports (lines 86-89).

Hereafter we provide our response and arguments to the referee’s comments:

  • Data included in our manuscript were obtained from HeLa cells, the cell line that we routinely use for the identification of proteins of interest with possible roles in repair. Contrary to the reviewer's report, we did not use HEK293T cells for intoxication assays. HEK293T were used to produce the viral particles for CRISPR/Cas9 depletion of copine- 1 and copine-3.
  • Our aim is clearly indicated in lines 79-81 (“Here, we seek to determine the proteomic repertoire of vesicles released by intoxicated cells during the repair of PM damage, with the aim to identify novel proteins involved in repair.”).
  • We do not consider that our study is a duplication of the Larpin et al 2020 and Wolfmeier et al. BBA 2016 papers.

In Larpin et al 2020, the authors assess the susceptibility of different immune cells to PLY, using a lymphoid cell line (Jurkat) and myeloid cell lines (U937 and THP-1). They showed that Jurkat were more sensitive due to their higher affinity for PLY and less efficient repair mechanisms. The authors claim that repair in myeloid cell lines is dependent on the expelling of microvesicles. In this study, microvesicles were not analyzed regarding their proteomic content, thus there is  no data duplication between this study and ours. Concerning the techniques used, the referee claims that we used the same techniques and experimental strategies that the authors use in  Larpin et al 2020, we want to stress that this is also the case for all the researchers working with extracellular vesicles in all fields. We all use the techniques available, and we don't really understand why this could be a problem.

In Wolfmeier et al 2016, the authors use a bunch of different cell lines (HEK293, SH-SY5Y, HeLa, etc) and report that cells are able to eliminate the PLY pores through the release of microvesicles, which they submit to mass spec for protein content determination. The rationale for the use of these different cells is missing. In this paper, the authors compare protein content of microvesicles with that of the “total cellular membrane preparations". Off note, this paper is properly cited in our manuscript and the data discussed (lines 487-491). Among the many differences between our work and the study published by Wolfmeier et al 2016 we want to highlight some: i) our study is a comparative study between two different CDCs, and we treated the mass spectrometry data in a way that bring out the commonalities between the repair processes triggered by the two toxins; ii) as control we used vesicles released by the non-intoxicated cells, with the aim to eliminate the proteins that are involved in the release of vesicles triggered by any kind of stimuli; iii) we provide data that validate our approach by showing for the first time that Copine-1 and Copine-3, enriched in EVs in response to PFT intoxication, are important factors for the repair of damage induced by PLY.

For all the reasons presented’, we disagree with the referee concerning the lack of novelty of our manuscript and her/his claim of data duplication. On the contrary, we believe that our study is a step forward in our understanding of PM repair in a more holistic view, without undermining the previous studies in the field. We are thus convinced that our study is of interest to researchers in the PFTs field and deserves to be published in Toxins.

Reviewer 2 Report

In this work, the authors collected extracellular vesicles released by HeLa cells exposed to two pore-forming toxins that belong to the family of the cholesterol-dependent cytolysins (CDCs): listeriolysin O (LLO) and pneumolysin (PLY). The overall idea is that proteomic analysis of the released vesicles may help identifying novel proteins involved in plasma membrane repair.

HeLa cells were exposed to sublytic concentrations of LLO and PLY which allow for plasma membrane resealing. Several mechanisms for the repair of cells injured by sublytic concentrations of CDCs have been proposed including the shedding of membrane vesicles. Therefore, it was a very clever idea to analyze the content of such vesicles to identify the machinery involved in plasma membrane repair. Such method could be applied to cells exposed to other families of pore-forming toxins and to other cellular contexts. The finding that the copines are repair proteins is novel and important.

Overall, I find this work interesting and impactful. The experimental work is rigorous and well-described. The article is well-written and clear. I have several comments which, if answered, could further improve this work.

1- The title is not fully appropriate. PLY and LLO are closely related pore-forming toxins that belong to the same family, bind to cholesterol, and use a similar fold and mechanism to perforate the PM. Overall, these two toxins display minor differences and don’t really qualify to be described as “different bacterial pore-forming toxins”. I strongly recommend modifying the title.

2- 116-117. Figure 1C shows that the non-intoxicated cell number increased during the 20-hour period, which reflects cell growth. However, cell numbers remained the same in toxin-exposed cells. This result is interpreted as “cells do not die but do not proliferate”. This is one possible explanation, but it is also possible that cell death compensates for cell growth, this should also be proposed.

3-It is well-known that plasma membrane repair is activated within seconds (or less) following damage. Therefore, plasma membrane machineries are repairing cells during the 15 min of toxin exposure. It is somewhat misleading to describe the recovery time as the events happening after toxin exposure while recovery started as soon as the toxins were added to the cell culture medium.

4-In the section of the proteomic analysis, it was not explained what criteria were used to categorize proteins as “potential contaminants”.

5-The proteomic of vesicles recovered from toxin-exposed cells is substantially remodeled, with 70 proteins enriched. These proteins include several families of proteins such as translational proteins, chromatin-binding or regulatory proteins, the significance of which was not really discussed.

6-Annexins and the ESCRT machinery have been involved in the shedding of vesicle during plasma membrane repair. The proteomic did not reveal enrichment in ESCRT associated protein and I wonder if the authors have any idea why this is the case.

7-The finding that copines may act as plasma membrane repair proteins is very interesting. As hypothesized by the authors, the machineries involved in plasma membrane repair are likely to be recovered in the extracellular vesicles, so it would be important to establish if the number and/or size of the extracellular vesicles is decreased in copine KO cells exposed to the toxins in comparison to control cells? This would indicate that the copines facilitate shedding of vesicles.

8-What are the time points of figure 6B, C? Are any annexins or AHNAK co-localizing with the copines?

Author Response

REVIEWER 2

We are thankful to the reviewer for her/his time and interest in our manuscript and for the issues  raised, which we believe allowed us to improve the quality of our manuscript. We provide below a point-by-point response to the raised questions and modified the manuscript according to the reviewer's suggestions.

1- The title is not fully appropriate. PLY and LLO are closely related pore-forming toxins that belong to the same family, bind to cholesterol, and use a similar fold and mechanism to perforate the PM. Overall, these two toxins display minor differences and don’t really qualify to be described as “different bacterial pore-forming toxins”. I strongly recommend modifying the title.

In agreement with the referee's suggestions we modified the title of our manuscript. The new proposed title is: “Cells responding to closely related cholesterol-dependent cytolysins release extracellular vesicles with a common proteomic content including membrane repair proteins”

2- 116-117. Figure 1C shows that the non-intoxicated cell number increased during the 20-hour period, which reflects cell growth. However, cell numbers remained the same in toxin-exposed cells. This result is interpreted as “cells do not die but do not proliferate”. This is one possible explanation, but it is also possible that cell death compensates for cell growth, this should also be proposed.

We understand the referee’s raised issue. However, several lines of evidence favor the hypothesis of cell survival and recovery instead of cell growth which would compensate for cell death. From our experience, when using lytic concentrations of PFTs, cell death occurs rapidly after intoxication and we already detect it at time 0. This is the case for cells intoxicated with 2 nM LLO or 0.5 nM of PLY, for which at time 0 we already have less cells due to the washout of dead cells during processing for flow cytometry analysis. Massive cell death affecting cell numbers  is less prone to take place after toxins washout. In addition, we did not detect any cell number changes in experiments using intermediate times of recovery (e.g. 2 h, 4 h, 6 h, etc). Finally, in time lapse microscopy experiments we never noticed cell death or detachment during intoxication with the used concentrations of PLY or after toxin washout.

3-It is well-known that plasma membrane repair is activated within seconds (or less) following damage. Therefore, plasma membrane machineries are repairing cells during the 15 min of toxin exposure. It is somewhat misleading to describe the recovery time as the events happening after toxin exposure while recovery started as soon as the toxins were added to the cell culture medium.

We understand the point raised by the referee. Indeed, during the 15 min of toxin exposure, the toxin binds to the cell plasma membrane, oligomerizes and forms pores. As soon as the plasma membrane is perforated some cells start to repair, however while the toxin is still present and free in the medium the damage is continuously occurring. It is difficult to synchronize damage. For these reasons, in our analysis we only consider time for recovery after toxin washout as no more damage can occur.

4-In the section of the proteomic analysis, it was not explained what criteria were used to categorize proteins as “potential contaminants”.

The software tool used for proteomics data analysis (MaxQuant version 1.6.9.0) comes with a small fasta file that includes common contaminants such as keratins, proteases used for (non-)tryptic digest, fluorescent probes such as YFP and others. The potential contaminants are therefore defined by the software tool and can be retrieved from the installation folder of MaxQuant or alternatively be accessed via this link http://www.coxdocs.org/doku.php?id=maxquant:start_downloads.htm. An explanatory sentence was added in the revised version of the manuscript (lines 349-351).

5-The proteomic of vesicles recovered from toxin-exposed cells is substantially remodeled, with 70 proteins enriched. These proteins include several families of proteins such as translational proteins, chromatin-binding or regulatory proteins, the significance of which was not really discussed.

Histones have been implicated in the response to PFTs, however not directly in repair. Several PFTs, including LLO and PLY, were shown to induce post-translational modifications of histones regulating gene expression (10.1073/pnas.0702729104). However, the timings and concentrations for intoxication are not the same as the one we used here. Whether histones detected in the extracellular vesicles show specific modifications (methylation, deacetylation, phosphorylation, etc) is unknown and needs to be investigated. Interestingly, some reports claim the existence of non-nuclear histones that can be secreted through the release of extracellular vesicles and that have a potential role in cell signaling and innate immunity (/10.1139/o06-082). Histones detected in the extracellular vesicles released by intoxicated cells may thus act as damage-associated molecular pattern signal instead of having an active role in repair of the plasma membrane damage (http://umu.diva-portal.org/smash/get/diva2:1203620/FULLTEXT03.pdf). This hypothesis sustains our view that released extracellular vesicles may act as a communication mechanism while also serving repair purposes.

Concerning the translational proteins that we also found enriched in the released vesicles, it is worth to remind that, in general, specific RNA molecules are packed into EVs to be secreted. We did not assess the presence of RNA in EVs in our study, however the detection of ribosome-associated proteins seems to suggest the presence of RNA, this will be investigated in the future. Ribosome-associated proteins were often detected in the EVs in other contexts (10.1038/s41598-018-28485-9; 10.1186/s12974-018-1204-7).

In our original version, we did not extend our discussion to all the protein families we found enriched in the vesicles. In the revised version of our manuscript, we added a small paragraph to broaden our discussion (lines 508-520).

6-Annexins and the ESCRT machinery have been involved in the shedding of vesicle during plasma membrane repair. The proteomic did not reveal enrichment in ESCRT associated protein and I wonder if the authors have any idea why this is the case.

Some of the ESCRT proteins, such as TSG101, CHMP4A-B, and VPS4B were reliably detected and quantified in the vesicles released by intoxicated cells (Supplementary table S1). However, these proteins did not show significantly different abundances in EVs from intoxicated and non-intoxicated cells (Supplementary tables S2, S3 and S4), which may suggest that ESCRT proteins are required for the shedding of EVs, regardless of the stimuli used. Comparative analysis such as ours, allow the identification of proteins involved in the secretion of EVs in a particular condition (10.1073/pnas.1521230113).

7-The finding that copines may act as plasma membrane repair proteins is very interesting. As hypothesized by the authors, the machineries involved in plasma membrane repair are likely to be recovered in the extracellular vesicles, so it would be important to establish if the number and/or size of the extracellular vesicles is decreased in copine KO cells exposed to the toxins in comparison to control cells? This would indicate that the copines facilitate shedding of vesicles.

We fully agree with the referee. In the future we plan to unravel the molecular role of copines in the plasma membrane repair of damage induced by pore-forming toxins. In this context, we will investigate whether the absence of copines (using the KO cells) would affect the release of extracellular vesicles, their size and their proteomic content.

8-What are the time points of figure 6B, C? Are any annexins or AHNAK co-localizing with the copines?

Cells shown in Fig 6B and C were intoxicated for 10 min, washed and immediately fixed (this info was added in the figure legend, line 462). We still don’t know whether copines localize with ANHAK or annexins. For the moment we only analyzed the colocalization with NMHCIIA accumulations, which are hallmarks of plasma membrane repair (10.15252/embr.201642833). The dynamics of copines recruitment to the plasma membrane needs to be followed by time lapse microscopy. In the future we plan to follow in real time the recruitment of copines together with the recruitment of annexins, NMHCIIA and AHNAK. This analysis should provide us information on the hierarchy of events taking place during the response to cholesterol-dependent cytolysins.

Reviewer 3 Report

This paper describes that bacterial pore-forming toxins (PFT) release extracellular vesicles with membrane repair proteins.

Shedding of extracellular vesicles (EVs) has been proposed as a key mechanism to eliminate PFT pores and restore PM integrity. The author showed that PFTs are partially eliminated through EVs release, and hyposize that  important proteins for PM repair might be included in EVs. 

Listeriolysisn O and Pneumolysin induced EV release, which is more and larger-EVs than non-intoxicated cells. They identified a cluster of 70 proteins including calcium-binding proteins and molecular chaperones. in EVs collected from intoxicated cells. Especially they identified Copine-a and Copine-3 are required for efficient repair of PFT-induced PM damage. 

The story about PFT and repair is interesting, and the various experiments were done well.

I recommend publishing in toxins.

Major points: 

1)What is the difference between extracellular vesicles and exosomes? Please explain.

2)Does the extracellular vesicles have a membrane?

3)The extracellular vesicles should have not only PFT but also usual cytoskeletal proteins and membrane proteins. But I think it is not necessary to include repair proteins in EVs. 

Why should EVs have repair proteins such as Copine-1 and Copine-3?

Is there any correlation between the shedding and repair?

4)Fig5

Please explain the difference between principal component 1 and component 2.

It would be necessary for general readers to toxins.

Minor point

Show scale bars in Fig 4A and B.

Author Response

REVIEWER 3

We are thankful to the reviewer for her/his interest in our manuscript and for the issues  raised. We provide below a point-by-point response to the raised questions and modified the manuscript according to the reviewer’s suggestions.

Major points:

1)What is the difference between extracellular vesicles and exosomes? Please explain.

Exosomes are a subtype of Extracellular vesicles (EVs). EV subtypes are differentiated by both their size and the nature of their biogenesis, though there is some overlap ultimately leading to some confusion about the nomenclature. Given the lack of specific markers for each of the aforementioned EV subpopulations, the International Society of Extracellular Vesicles (ISEV) has suggested the generic term “EVs” for the vesicles released from the cell (for details please see 10.3389/fphys.2020.604274; 10.1080/20013078.2018.1535750; https://doi.org/10.1080/20013078.2019.1648167).

Exosomes are considered the smallest EVs and are produced in the endosomal compartment of eukaryotic cells through the inward budding of the endosomal membrane, leading to the formation of multivesicular bodies that later fuse with the plasma membrane to release their content. The EVs that we mention in our manuscript are larger in size and most likely are formed by outward budding and pinched off from the plasma membrane.

2) Do the extracellular vesicles have a membrane?

Yes, extracellular vesicles are membrane bound vesicles that are secreted by cells into the extracellular space ( doi:10.3402/jev.v4.27066; doi:10.3390/cells8070727).

3)The extracellular vesicles should have not only PFT but also usual cytoskeletal proteins and membrane proteins. But I think it is not necessary to include repair proteins in EVs.

Why should EVs have repair proteins such as Copine-1 and Copine-3?

Is there any correlation between the shedding and repair?

We understand the point raised by the referee. Indeed, one could expect that the repair proteins are not required to be present or enriched on the released vesicles. However, we can also put forward the possibility that, because they accumulate in the vicinity of the damaged site, where presumably blebbing is occurring and vesicles are being released, repair proteins are packed into the vesicles. We favored this second possibility and hypothesized that at least part of the proteins involved in repair would be included in the vesicles during the pinching off and the release. Our data confirm this hypothesis as proteins already characterized in plasma membrane repair processes appeared with increased abundance in the vesicles released by intoxicated cells.

Yes, shedding and repair are correlated. For efficient repair shedding needs to occur in a tightly regulated manner (10.15252/embr.201642833).

4)Fig5

Please explain the difference between principal component 1 and component 2.

It would be necessary for general readers to toxins.

Principal component analysis (PCA) is a statistical analysis technique utilized in many fields including proteomics research. It is an intuitive method that reduces a large number of variables to a smaller number of groups that can be more readily visualized and understood. PCA is a way to bring out strong patterns from large and complex datasets. Principal components are new variables that are constructed as linear combinations of the initial variables. These combinations are done in such a way that the new variables (i.e., principal components) are uncorrelated and most of the information within the initial variables is squeezed or compressed into the first components. From our data, principal component 1 and principal component 2 explain 38.1 % and 14.5 % of the total variance, respectively. PC1 is thus the variable that mostly explains the variance and PC2 is the second variable that better explains the variance.

An explanatory sentence was added in lines 323 and 324.

Minor point

Show scale bars in Fig 4A and B.

Scale bars are shown in Figs 4A and B and their corresponding value in nm is now indicated in the figure legend (line 298).